# Effects of Elevation and Distance from Highway on the Abundance and Community Structure of Bacteria in Soil along Qinghai-Tibet Highway

**DOI:** 10.3390/ijerph182413137

**Published:** 2021-12-13

**Authors:** Zhuocheng Liu, Yangang Yang, Shuangxuan Ji, Di Dong, Yinruizhi Li, Mengdi Wang, Liebao Han, Xueping Chen

**Affiliations:** 1School of Grassland Science, Beijing Forestry University, Beijing 100083, China; liuzhuocheng@bjfu.edu.cn (Z.L.); jsx11223345@163.com (S.J.); didoscori@163.com (D.D.); liyinruizhi@163.com (Y.L.); mengdi0627@163.com (M.W.); 2Environmental Protection and Soil and Water Conservation Research Center, China Academy of Transportation Sciences, Beijing 100029, China; Ecologyoung@126.com

**Keywords:** Qinghai-Tibet Plateau, plant community structure, bacterial community structure, heavy metals, Tibet Highway

## Abstract

In recent years, highway construction in the Qinghai-Tibet Plateau (QTP) has developed rapidly. When the highway passes through grassland, the soil, vegetation, and ecological environment along the line are disturbed. However, the impact on soil bacteria is still unclear. Soil bacteria play an important role in the ecological environment. The Qinghai-Tibet Highway (QTH) was selected as the research object to explore the changes in bacterial community structure, vegetation, soil, and other indicators. The results showed that the highway-related activities increased the degradation of vegetation along the road, significantly changed the physical and chemical properties of soil, and caused heavy metal pollution. These environmental factors affected the diversity and community structure of soil bacteria. This kind of disturbance shows a trend of gradually increasing from near to far from the highway. *Gemmatimonas*, *Terrimonas*, *Nitrospira* and *Bacillus* are more tolerant to environmental changes along the highway, while *Barnesiella*, and *Blastococcus* are more sensitive. The content of nitrate decreased and the content of ammonium nitrogen increased in the disturbed area, increasing the abundance of nitrifying bacteria. Therefore, the main factor of the disturbance of the QTH on the grassland is the decline of soil nutrient content, and the supplement of soil nutrients such as carbon and nitrogen should be taken into account in the process of ecological restoration of grassland along the line.

## 1. Introduction

As the highest geographical unit in the world, the Qinghai-Tibet Plateau (QTP) has a particular and representative ecological environment. The QTP is of great significance to global climate change, carbon cycle, and biological germplasm resources [1,2,3,4]. As a fragile ecosystem, the QTP ecosystem is particularly vulnerable to human activities [5,6]. In recent years, with the rapid economic development of the Qinghai-Tibet region, road transportation has also continued to develop [7,8]. The total length of highways in the Tibet Autonomous Region increased from 15,852 km in 1978 to 117,000 km in 2021 [9]. Highway traffic inevitably disturbs natural grasslands and has a series of negative influences on the ecosystem, including soil erosion, vegetation destruction, and water quality deterioration [10,11,12,13]. The sharp increase in negative impacts has resulted in imbalances in the self-regulation of the grassland ecosystem, and varying degrees of degradation of grasslands along highways [14].

As an important part of the ecosystem, soil microorganisms participate in the degradation of organic matter, biogeochemical cycles, and the maintenance of soil structure and are closely related to the degradation of grassland ecosystems [15]. In natural grassland, vegetation growth, soil nutrients, moisture, pH value, etc. determine the community composition and structure of soil microorganisms [16,17]. In addition, it changes with depth due to factors such as redox conditions and soil nutrients. In the QTP, soil microorganisms are mainly distributed in the topsoil (0–20 cm) [18]. Many scholars point out that soil bacterial communities are extremely sensitive to human activities [19]. However, the response of soil bacterial communities to environmental changes resulting from road traffic is currently unclear.

Road traffic and construction will produce different types of heavy metal pollutants [20,21]. The components, fuels, and lubricants of vehicles and paving materials may contain heavy metals such as copper, copper, lead, zinc, and cadmium [22]. The burning of liquid fuels, the use of lubricating oil, the wear of vehicles and the loss of pavement will bring in heavy metal ion particles. These particles enter the ecosystem on both sides of the road, possibly through sedimentation pavement runoff and splashing [23]. Studies have shown that the concentrations of heavy metals in soil were significantly affected by road construction and traffic. The degree of impact is inversely proportional to the distance from the road [24,25]. Many studies have shown that the deposition of heavy metal pollutants in soil will affect the composition and structure of soil bacterial communities [26,27]. Additionally, soil nutrient content is another important factor affecting soil bacterial communities. Road construction and traffic drainage will cause varying degrees of soil erosion, topsoil stripping, and other changes. Many studies have shown that soil erosion can change physical and chemical properties such as soil density, soil moisture, and pH, leading to soil nutrient loss [28,29]. As the mediator of more than 90% of the energy and material exchange in the soil ecosystem, the bacterial community structure is bound to be affected to varying degrees [30].

In general, exploration of the influences of soil nutrient changes and heavy metal on the structure of soil bacteria communities can provide data support for grassland ecological restoration along the highway, and can also provide a decision for road construction in the QTP. The Qinghai-Tibet Highway (QTH) provides favorable conditions for exploring the relationship between the highway and soil microorganisms [25]. The grassland ecosystems in most areas along the route are well preserved, are very sensitive to external influences, and have poor natural recovery capabilities [21]. As an uninhabited area and nature reserve, there is almost no human disturbance in this area except for traffic [31,32]. The Golmud-Lhasa section of the QTH passes through areas of different altitudes, climates, soil types, and vegetation types. We assume that the QTH has an impact on the structure of the grassland soil bacterial community by changing soil and heavy metals factors. In this study, treatments at different distances from the highway and altitude were selected, and the bacterial communities along the QTH were determined by high-throughput sequencing technology. The results showed that: (1) The soil bacterial community structure changes with the distance from the QTH; (2) In high altitude areas, the soil bacterial community is more susceptible to highway impacts; (3) Soil nutrients are the main driving factor of soil bacterial community structure change; (4) Within a certain distance from the QTH, the heavy metal content was correlated with soil bacterial community diversity.

## 2. Materials and Methods

### 2.1. Study Area and Sampling Lines

This study selects the QTH (G109 National Highway) Golmud to Lhasa section as the research object. The region where the QTH passes through has a temperate/sub-frigid continental climate, with long and cold winters, strong winds, little rain in summer, and short spring and autumn. This section of the highway crosses two major soil types: alpine prairie soils (Cryuborolls) and alpine meadow soils (Cryaquet) and a small portion of gray-brown desert soils (Gypsic Haplosalid). In this area, the highway passes through the Hoh Xil National Nature Reserve and the Sanjiangyuan Nature Reserve. There is no large-scale agriculture, animal husbandry, and manufacturing industries, and the natural grassland is less affected except for the highway. The effect of traffic activity on the deposition of soil pollutants was stronger closer to the road and weakened with the increase of distance from the road [25,32]. Therefore, a plot 400 m from the highway was collected as a control in each area that may not be affected by traffic activity.

More than 91% of the studied sections have an altitude of >4000 m, and the highest altitude is 5231 m, which is located at Tanggula Mountain Pass. In this study, 3 sampling zones at 4000 m, 4600 m, and 5231 m were selected, and one sampling zone was set in an area within 400 m of each altitude where there were no other human influence factors except roads (sampling zone 35°47′ N, 94°20′ E; sampling line 2 at 34°27′ N, 92°44′ E; sampling zone 3 at 32°53′ N, 91°55′ E). The sampling lines were perpendicular to the highway and is 400 m long and 10 m wide (Figure 1). The altitude drop in the sampling zone is less than 30 m. For each sampling zone, 4 plots were set at a distance of 5 m, 20 m, 50 m, and 100 m away from the highway shoulder, and 1 plot was set at a distance of 400 m from the shoulder, which was used as a control plot. Four samples of 1 m × 1 m are randomly set for each plot.

### 2.2. Vegetation Samples

In each plot, we investigated plant species, abundance, height, coverage and other indicators, and calculate important values. Species diversity is measured by species richness index and Shannon diversity index. All the above-ground parts of the sample were collected to determine the above-ground biomass.

### 2.3. Soil Sampling

July is the growing season for plateau vegetation. The middle of July 2019 was selected as the time of soil sample collection. Use a thermometer to measure the soil temperature before sampling. The soil density was determined by a soil compaction meter (TJSD-7500II). Soil samples were collected using a soil auger with a diameter of 5 cm and a sampling depth of 0–20 cm. Ten soil cores were evenly distributed in each quadrat. Five plots were set for each site. Four quadrats were set for each plot for repetition, and 5 times of sampling were set in each quadrat for mixing. Thus, a total of 15 treatments were set and 60 soil samples were collected.

After removing the rocks and grass roots, the sample is thoroughly mixed and passed through a 2 mm sieve. Approximately 5 g of samples were immediately put in a 2 mL centrifuge tube and brought back at low temperature, and kept in a refrigerator at −80 °C for DNA analysis. About 300 g samples were sealed and brought back into the refrigerator at 4 °C for soil physical and chemical analysis. About 1000 g of Soil samples are dried for soil nutrient and heavy metal analysis. During the entire collection and processing of soil samples, bacteria and heavy metal contamination should be avoided.

### 2.4. Soil Laboratory Analysis

#### Chemical Analysis

Soil pH was determined by mixing the soil sample and water in a ratio of 1:5 using a pH meter. Soil organic C content was determined by a potassium dichromate external heating method [33]. Soil total N was determined by Kjeldahl digestion and automatic azotometer [34]. Soil alkali-hydrolytic nitrogen, total nitrogen, available P, and available K were determined by the method of previous studies [35,36].

The contents of Cu and Zn in the soil were determined by flame atomic absorption spectrophotometry. The main instrument is a TAS-990F atomic absorption spectrophotometer. Soil Pb and Cd contents were determined by graphite furnace atomic absorption spectrophotometry using a 240ZAA atomic absorption spectrophotometer [37].

### 2.5. DNA Extraction, PCR Amplification

According to the manufacturer’s instructions, use the Fast DNA SPIN Kit for Soil (DNeasy PowerSoil Kit, QIAGEN, Hilden, Germany) to extract total soil DNA. Finally, the DNA was eluted with 100 μL DNA eluent in the kit. Dilute the successfully extracted DNA to a concentration of 1 ng/μL and store at −20 °C until further processing.

The barcoded primers and Takara Ex Taq (Takara) were used to amplify the *16S rRNA* genes of bacteria using the diluted DNA as a template, and V3-V4 variable regions of *16S rRNA* genes were amplified with universal primers 343F and 798R for bacterial diversity analysis [38]. To verify the size and quality of the PCR products, all of them were electrophoresed in 1.5% (wt/vol) agarose [39,40].

### 2.6. Cloning, Sequencing and Phylogenetic Analysis

The quality of the amplicons was visualized using gel electrophoresis, purified with AMPure XP beads (Agencourt), and subjected to another round of PCR amplification. After purification again using AM-Pure XP magnetic beads, the final amplicons were quantified using the Qubit dsDNA Detection Kit. Equal amounts of purified amplicons were pooled for subsequent sequencing [41,42].

The raw sequencing data were in FASTQ format [43]. Pre-processing of double-ended reads, including detection and cleavage of ambiguous bases (N), was performed using Trimmomatic software (Bolger AM: Golm, Brandenburg, Germany) [44]. Low-quality sequences with an average quality score below 20 were cut off using the sliding window pruning method [40]. Parameters of assembly were: 10 bp of minimal overlapping, 200 bp of maximum overlapping, and 20% of maximum mismatch rate. Assembly parameters were: minimum overlap of 10 bp, maximum overlap of 200 bp, and maximum mismatch of 20%.Reads with 75% of bases above Q20 were retained. Reads with chimeras were then detected and deleted. QIIME software (version 1.8.0, Caporaso JG: Boulder, CO, USA) was used to implement the above two steps [45].

Using the Vsearch software (Edgar RC: Cambridge, MD, USA) with 97% similarity threshold, the clean readings were generated by primer sequencing and clustering to generate surgical taxons (OTUs) [46]. We used the QIIME package to choose a representative reading for each OTU; the RDP classifier (confidence threshold 70%) to annotate all representative readings and annotate the Silva database version 123 (*16s rDNA*) [42]; and the blast to annotate all representative reads and blast the Unite database (ITSs rDNA) [43].

### 2.7. Statistical Analyses

Statistical analysis software such as Excel 2010 (Microsoft: Seattle, WA, USA) and SPSS 22 (International Business Machines Corporation: Armonk, NY, USA) were used to arrange and plot the measured data. The data were analyzed by Microsoft Excel 2010, and the differences of vegetation data, soil physical, and chemical properties and soil bacterial diversity index were analyzed by One-way ANOVA in SPSS 22. R 3.5.2 was used to perform analyses of species composition and diversity, non-metric multidimensional scaling (NMDS), Adonis and Mantel tests. The OmicShare tool was used to perform the FAPROTAX analysis (https://www.omicshare.com/tools, accessed on 10 September 2021).

## 3. Results

### 3.1. Environmental Factors

The coverage in the area within 100 m from the highway reduced more significantly than the control (400 m) at three sites (Figure 2B). The coverage at 20 m decreased by 64.44% (0.20), 67.24% (0.24) and 64.86% (0.33) in 4000 m, 4600 m and 5200 m sites, respectively. The aboveground biomass was negatively correlated with distance in three sites. The aboveground biomass decreased significantly within 100 m, 50 m, and 100 m from the road, respectively (Figure 2A). The plant Shannon index decreased significantly in 4000 m and 5200 m sites within 20 m from the highway and decreased significantly at the 4600 m site within 50 m from the highway. For the plots 20 m away from the highway, the values decreased by 28.14% (1.62), 14.05% (1.56), and 28.80% (1.21) compared with the control (400 m) in 4000 m, 4600 m, and 5200 m sites, respectively (Figure 2C). The Simpson index (Figure 2D) of plants also shows similar changes in the areas close to the road (50 m, 20 m, and 5 m plots). Compared with the control (400 m), the plants’ heights were significantly higher. The increase ranges were 561.17% (35.59 cm), 250.46% (16.15 cm) and 59.38% (8.54 cm).

The soil moisture (SM) was significantly reduced by 58.63%, 38.45%, and 73.04% within 100 m along the highway at 20 m plots from the highway in 4000 m, 4600 m, and 5200 m sites, respectively (Figure 3A). In the 5200 m site, soil pH significantly decreased by 6.26%, 6.07%, 3.47%, and 2.11% at 5 m, 20 m, 50 m, and 100 m from the road, respectively (Figure 3B). At the same time, smaller changes occurred at 4000 m and 4600 m, with only significant decreases of 2.75% and 1.29% at 5 m from the road, respectively. The soil temperature in the areas 100 m and 50 m away from the highway decreased significantly compared with the control, 4000 m site and 5200 m site, respectively (Figure 3C). Soil compaction only changed significantly at 5200 m and decreased by 9. 14% (1715.75 Pa) and 5.69% (1780.75 Pa) at 5 m and 20 m away from the road, respectively (Figure 3D).

SOC is negatively related to the distance from the road. This trend is most obvious at 5200 m (Figure 4A). Roads significantly reduced soil organic carbon by 33.85% (.84 g·kg^−1^), 36.43% (8.49 g·kg^−1^), and 32.72% (8.99 g·kg^−1^) at plots 5 m, 50 m, and 100 m away from the road, respectively. Soil organic carbon content decreased by 70.19% (8.35 g·kg^−1^) and 66.01% (9.52 g·kg^−1^) at 5 m and 20 m plots from the road, respectively, compared with the control (400 m) at 5200 m. Soil total nitrogen content decreased by 67. 48%, 60.98%, 19.86% and 21.12% at 5 m, 20 m, 50 m, and 100 m away from the highway, respectively (Figure 4B). In the 4600 m site, soil total nitrogen content significantly decreased by 27.17% at 20 m from the road. At the same time, the change of soil total nitrogen content was not significant at the 4000 m site. The change trend of alkali-hydrolyzable nitrogen (AN) was the same as TN in the 5200 m site (Figure 4C). The difference is that angle in which the content of soil AN decreased significantly became within 50 m from the road. On the contrary, at the site of 4600 m, the content of soil AN at 5 m plot away from the road was significantly increased by 16.36% (67.02 mg·kg^−1^) compared with the control (400 m). At the same time, at the site of 4000 m, the content of soil AP (Figure 4D) at the plot of 20 m from the road was significantly increased by 30.34% (80.00 mg·kg^−1^) compared with the control (400 m). In 5200 m site, the content of soil available phosphorus increased by 119.04% (4.80 mg·kg^−1^) and 198.97% (6.56 mg·kg^−1^) at 5 m and 20 m distance respectively. The content of soil available phosphorus increased at the distance of 50 m (56.15%) and 20 m (169.42%) from the highway, and at the distance of 4000 m and 4600 m, respectively. The change of soil AK with distance was not significant (Figure 4E).

The Contents of Cu, Zn, and Cd increased significantly in the 4000 m site, and the increasing areas were mainly concentrated in the area 20 m away from the road (Figure 5). The three heavy metals contents increased by 17.18%, 14.73%, and 31.58% at 20 m from the road, compared with the control(400 m in 4000 m sites).The total lead content in the soil increased by 41.92% and 63.57% at 5 m and 20 m, respectively, in the 4600 m sites.

### 3.2. Bacterial α-Diversity

Alpha Diversity Index is used to reflect species diversity. Observed Species and Chao1 are used to indicate the actual number of OTUs in the community. The larger the Shannon Wiener and Simpson values are, the higher the community diversity is and the more uniform the individual distribution is. The Good’s Coverage index reflects the sequencing depth, and the closer the index is to 1, it indicates that the sequencing depth has covered all species in the sample. The larger the value of the Phylogenetic Diversity index, indicates that the species that make up the biome are further apart in evolution. Specaccum accumulation curves determine whether the amount of sequenced data completely covers the species on the total sample. After pruning and quality filtering, 98,532 high-quality readings were obtained from a total of 60 samples from three altitudes, and the amount of clean tags data after quality control ranged from 4802 to 52,373. The data size of the valid tags (used for analysis) ranged from 11,648 to 42,192, with an average length of 426.58 to 432.68 bp.

The number of OTUs obtained from all treated soil samples ranged from 786 to 1987, and the species coverage (Good’s Coverage index) was more than 90%. Based on a genetic distance of 3%, the sparse curve tends to saturate the plateau, indicating that the sequencing depth is sufficient (Figure 6D). The species accumulation curve showed that the sample size was sufficient for subsequent data analysis (Figure 6E).

The impact of highway traffic on bacterial α-diversity was mainly reflected in the 5200 m site, and the OTUs of soil bacteria significantly increased by 12.78% (1713) and 11.67% (1696) in the 5 m and 20 m plots in the 5200 m sites, respectively. Consistently, the bacterial Shannon index also increased significantly by 4.40% (9.40) and 3.86% (9.35) at 5 m and 20 m from the road, respectively, at 5200 m. In the 4600 m site, the bacterial CHAO1 index significantly increased by 8.93% (2901.93) at the 5 m site. On the contrary, the number of bacterial OTUs and Shannon index decreased significantly by 13.35% (1544.7) and 7.61% (8.56), respectively at 50 m from the road in 4000 m site (Figure 6).

Pearson correlation analysis showed that the environmental factors, especially altitude, SOC, AK, TN, AN, SM, pH, Density, Cu, and Simpson were significantly correlated with the bacterial α diversity index. OUT number and CHAO were negatively correlated with the PD whole tree index. In addition, there was a significant positive correlation between altitude and bacterial Simpson index. Cu content was significantly negatively correlated with Simpson index and Good’s Coverage index. The Simpson index of plants was negatively correlated with that of bacteria (Figure 6D).

### 3.3. Bacterial Community Structure

The relative abundance of phylum-level bacterial communities is shown in Figure 7. The major phyla in all samples were *Proteobacteria* (20.83–40.02%), *Actinobacteria* (15.28–31.09%), *Bacteroidetes* (13.16–27.99%), *Firmicutes* (9.72–27.99%), *Gemmatimonadetes* (1.61–8.10%), *Acidobacteria* (1.39–5.06%), *Nitrospirae* (0.15–1.13%), *Chloroflexi* (0.13–0.86%), *Cyanobacteria* (0.15–1.08%) and *Chlorobi* (0.13–0.47%), with a total of more than 98% in each sample(Appendix A).

The *Proteobacteria* abundance decreased by 30.65% (28.03%) at the 50 m site than the control (400 m). The *Actinomycetes* abundance significantly decreased by 47.07% (20 m plot, 15.28%) in the 4000 m and 4600 m sites than the control. Conversely, the abundance of *Actinomycetes* significantly increased by 87.71% (29.03%) and 98.43% (28.03%) at 5 m and 20 m, respectively. *Bacteroides* abundance in the 20 m plot significantly increased by 75.92% (23.15%) in the 4000 m site compared with the control, while in the 5 m plot it significantly decreased by 34.82% (19.08%) in the 5200 m site compared with the control. The same results occurred with acid bacilli. At the 4000 m and 4600 m sites in the highway disturbance area, the abundance of acid bacilli at the 20 m and 50 m plots showed a significant decrease than control. However, the abundance of *Acidobacteria* significantly increased by 88.16% (2.61%) and 138.27% (3.31%) in the 5 m and 100 m sites, respectively. In addition, *Firmicutes* abundance in the 5 m and 20 m plots decreased significantly by 59.50% (10.61%) and 42.23% (15.13%) compared with the control in the 5200 m site. The abundance of *Nitrospirillum*, *Chlorospirillum*, *Cyanobacteria* and *Chlorobacteria* increased significantly in different degrees in the highway disturbance area.

In the genus level, the abundance of bacteria was analyzed by LEfSe in each site at each elevation, and the results are shown in Figure 8. Road traffic significantly increased the number of 4, 8, and 14 taxa of soil bacterial communities within 100 m (4000 m, 4600 m, and 5200 m), and significantly decreased the number of 5, 2, and 4 taxa of soil bacterial communities. *Flavisolibacter*, *Gemmatimonas*, *Microvirga*, *Nocardioides,* and *Rubrobacter* are enriched at sites of 4600 m and 5200 m in the disturbed area of the highway. *Crossiella* and *Gaiella* are enriched in the disturbed area at 4000 m and 5200 m sites respectively. At the same time, *Barnesiella* and *Blastococcus* were the main bacterial genera enriched in the unaffected area. *Flavisolibacter* abundance increased significantly at 5 m and 20 m sites at 4600 m and 5200 m sites, respectively. In addition, *Flavisolibacter* is also the genus with the highest importance index in 5200 m site in the random forest analysis. *Alistipes* and *Barnesiella* increased in abundance in the undisturbed area at the 5200 m site. In the 4600 m site, *Gemmatimonas*, *Rubrobacter*, and *Microvirga* all increased in abundance close to the road, consistent with the 5200 m site. *Pseudomonas* was the most important genus in random forest analysis at the 4000 m site, and its abundance increased significantly at the 20 m site, and the content of soil alkali-hydrolysable nitrogen also increased significantly at the 20 m site. *Marmoricola* abundance was lower in the highway disturbance area compared to the control in the 4000 m site. *Nocardioides* were enriched at 5 m in 4600 m and 5200 m sites, in contrast to 4000 m in the control (400 m) group. In random forest analysis, the top 10 genera were *Skermanella*, *Prevotella* 9, *Rubrobacter*, *Gaiella*, *Blastococcus*, *Crossiella*, *Flavisolibacter*, *Faecalibacterium*, *Pseudomonas,* and *Microvirga* (Figure 8A).

### 3.4. Bacterial β-Diversity

ADONIS analysis showed significant differences in vegetation and bacterial community structure between distances at the same elevation, as well as between elevations (Appendix A). From the results of NMDS analysis(Figure 9), it can be seen that there are significant differences in the structure of soil bacterial community among the samples at different altitudes (Stress value is 0.084), and the effect of altitude on bacterial community structure was more significant than that of distance. In different regions, significant differences in soil bacterial community structure were revealed among different distances (Stress values were 0. 058, 0.045, and 0. 043), which indicated that the soil bacterial community structure in the highway disturbed zone had changed significantly.

According to the RDA results, soil bacterial community structures in three sites could be distinguished by plant and soil factors (Figure 10). In the site of 4000 m, AN had the greatest impact on the bacterial community. The first and second axes explained 54.70% and 18.53% of the variation, respectively. AK, SOC, and Zn are the main correlation factors of axis 1. Associated with axis 2 are mainly AN and AK (Figure 10A). In the 4600 m site, the most important factor is distance, followed by TN and coverage (Figure 10B). The first axis (mainly related to distance and TN) and the second axis (mainly related to SM and AP) described 35.70% and 33.37% of the variation, respectively. In the 5200 m site, SOC and pH had the greatest impact on the bacterial community, followed by distance (Figure 10C). The first axis, mainly related to SOC, pH, and AN explained 60.96% of the variation. At that same time, the second axis was mainly related to Cd and AN explained 9.95% of the variation.

### 3.5. Relative Effects of Environmental Factors on Bacterial Communities

Pearson correlation coefficient analysis results between bacterial abundance and environmental factors are shown in Figure 11. At the phylum level, the abundance of *Proteobacteria* and *Fibrobacteres* was positively correlated with AK, SM, pH, and Cu (*p* < 0.01), and negatively correlated with SOC and TN (*p* < 0.05). *Actinobacteria* abundance was positively correlated with AK, SM, Zn, and Cu (*p* < 0.05). *Bacteroidetes* were negatively correlated with AK, SM, pH, Zn, and Cu (*p* < 0.05) and positively correlated with SOC, TN, and AN (*p* < 0.01). At the genus level, the abundance of *Lactobacillus* and *Enterococcus* was positively correlated with Cu, Zn, pH, and SM (*p* < 0.01), and negatively correlated with SOC (*p* < 0.05). The abundance of *Sphingomonas* and *Rubrobacter* was positively correlated with AK, Cu, pH, and SM (*p* < 0.001), and negatively correlated with SOC, TN, Density, and AN (*p* < 0.01). *Faecalibacterium*, *Prevotella* 9, and *Blautia* were negatively correlated with AK, SM, pH, and Cu (*p* < 0.01), and positively correlated with SOC and TN (*p* < 0.001). The abundance of *Bacteroides* was negatively correlated with AK, Zn, pH, and SM (*p* < 0.05). In addition, *Gemmatimonas* was positively correlated with Zn (*p* < 0.05) and negatively correlated with SOC (*p* < 0.01). These results show that soil properties, especially SOC, TN, AN, AK, SM, pH, Cu, and Zn have significant effects on soil bacterial communities.

### 3.6. Bacterial Functional Community

Effect of road traffic on soil bacterial function FAPROTAX function prediction results (Figure 12) showed that chemoheterotrophy, fermentation, animal parasites or symbionts, and nitrate reduction are the main functional groups along the highway. Among the top 30 functional predicted relative abundances, 25 species showed significant differences (*p* < 0.05): chemoheterotrophy, human pathogens, predatory or exoparasitic, aerobic ammonia oxidation, ureolysis, cellulolysis. The functional groups of ligninolysis, sulfate respiration and aromatic compound degradation increased in the disturbed area. The functional groups showing a downward trend in the highway disturbance area included fermentation, animal symbionts, photoheterotrophy, chlorate reducers and other functional groups. In addition, soil nitrifying bacteria such as ammonia oxidation, nitrite oxidation and nitrification showed an increasing trend in the disturbed area from the road compared with the control. Nitrate reduction and nitrogen fixation related to soil denitrification were significantly decreased in the sites within 100 m from the road.

### 3.7. Variation Partitioning

The results of variation partitioning analysis (VPA) showed that environmental factors such as altitude, vegetation, soil, and heavy metals could explain 32.71% of the variation in soil bacterial community structure, and the unexplained rate was 67.29%, of which the interaction of soil, vegetation, and heavy metals could explain 12.42% (Figure 13A). Environmental factors such as altitude, vegetation, soil, and heavy metals accounted for 34.89% and 65.11% of the changes in the ecological functional structure of soil bacteria FAPROTAX, of which the soil factor alone accounted for 8.02% and the interaction of soil, vegetation, and heavy metals accounted for 14.76% (Figure 13B). Compared to the soil bacterial community structure, the coupling effect of soil and vegetation had a greater impact on the functional structure of bacteria.

## 4. Discussion

### 4.1. Impact of Highway on Plant and Soil Physical and Chemical Properties

During the highway construction and operation, different degrees of impact were inevitable on the grassland soil environment and vegetation along the highway. As we can predict, the grassland vegetation in the areas close to the highway has been degraded to varying degrees. In general, results showed the closer to the highway, the lower the coverage and diversity and biomass of grassland vegetation were. This is consistent with many previous research results [47]. This damage includes the direct excavation and rolling of grassland vegetation by construction and access vehicles [11]. It may also be indirectly caused by soil erosion caused by the change of terrain on both sides of the highway [12,48]. In this study, the vegetation height was higher in areas closer to the road. In the field investigation, we found that this was because some invasive plants such as *Stipa purpurea* appeared after the original low grassland of the QTP was disturbed and degraded [49]. At the same time, heavy metal pollution caused by road traffic has further aggravated the degradation of grassland vegetation [50]. In different altitude environments, the degradation of grassland vegetation is also different, of which 4600 m is similar to 5200 m, and the degradation is most obvious in the 4000 m environment. There may be two reasons, one is that the 4000 m section is closer to the city (Golmud), and the traffic flow is larger. On the other hand, because the vegetation type of the 4000 m site is desert grassland, the vegetation diversity is low, and the ecological environment is fragile. It is more vulnerable to the interference of environmental factors.

Consistent with previous studies, our results indicate that highway slopes affect soil nutrient content, resulting in lower pH. Our results show that the soil moisture decreased significantly in the highway disturbed area. Engineering disturbance may cause soil compaction or decomposition by destroying original vegetation, altering soil porosity, and disrupting soil aggregates [51,52,53]. At the same time, the destruction of vegetation in the disturbed area of the highway leads to the decrease of the fixation ability of roots to soil, and the process of soil erosion may cause the loosening of soil structure. Our results show that soil pH is sensitive to engineering disturbance, which may be related to soil parent material [54,55]. Road disturbance affects soil pH by destroying native vegetation and affecting soil pH and nitrogen levels in plant material [56]. In addition, increased soil acidification may lead to increased leaching of cationic nutrients from the soil, thereby exacerbating the deficiency of certain nutrients essential for plant growth and ultimately leading to reduced plant productivity [57]. Because of the decrease of plant litter, the activity of soil denitrifying bacteria was promoted, and the mineralized nitrogen tended to nitrify, which led to the decrease of soil pH [56]. The decrease of soil moisture content (SM) in the highway disturbance area is related to the coverage of vegetation [53,58]. A reduction in the capacity of the soil to retain water due to degradation of vegetation. The change of soil temperature is contrary to the results of previous studies, which may be related to the weather at the time of sampling, or because the change of vegetation cover affects the thermal insulation performance. This requires further study.

Our results show that the soil SOC, TN, and AK in the disturbed areas of the highway do not change significantly compared to the control areas at 4000 m and 4600 m sites. The research of Pan and Jiang on highway slopes also shows the same situation [53,59]. All study areas are located below 4000 m and include the Tibetan Plateau region. In contrast, at the 5200 m site, the SOC, TN, and AK of highway disturbed area decreased significantly in our study, suggesting that SOC, TN, and AK may be more sensitive to the impact of the highway at higher altitudes. He et al.’s study showed that SOC and TN of highway disturbed areas significantly reduced compared with control [14]. The results of this study are consistent with this. However, there are few studies on the soil SOC and N contents in the highway disturbed area at an altitude of more than 5000 m. Therefore, we analyzed the differences of soil nitrate and ammonium nitrogen contents and related bacterial abundances among the three altitudes in this study. Our study revealed that due to the low oxygen content in high altitude areas, a low-oxygen environment is more conducive to denitrification, resulting in significantly higher soil ammonium nitrogen content and significantly lower nitrate content at other altitudes (Figure 14A,B). At the same time, the contents of TN and MBN and the abundance of Nitrospirillum in the soil at the altitude of 5200 m were significantly higher than those at other altitudes (Figure 14C,D), possibly due to the differences in biomass and coverage of grasslands and human disturbance. This also led to a significantly higher abundance of bacteria associated with the soil nitrogen cycle, including nitrifying and denitrifying bacteria. This indicates that the soil nitrogen content is higher and the activity of related bacteria is more vigorous at the altitude of 5200 m, which may cause the soil N indicators more sensitive to the impact of the highway at high altitudes. The abundance of nitrifying bacteria is proportional to that of denitrifying bacteria. This result is in accordance with the law expounded by the predecessors [60].

The increase of soil available phosphorus may be affected by many factors. In alkaline soils, the content of calcium ions is usually high in alkaline or calcareous soils, and phosphate ions are easy to form calcium phosphate precipitation with calcium ions, thus reducing the content of available phosphorus. In this study, the pH value of the road disturbance area decreased, which promoted the dissolution of the P element in alkaline soil into AP and increased soil AP. Simultaneously, the abundance of some bacteria (such as *Bradyrhizobium* and *Mycobacterium*) related to phosphorus dissolution increased in the disturbed area of the highway, which may also be the reason for the significant increase of available phosphorus content. Ma et al. (2013) recorded relatively rich AP levels on the slopes of the Beijing-Chengde Phase III Expressway [61], consistent with the findings of this research. The ratio of carbon to nitrogen increased significantly in the disturbed zone.

Many heavy metals have been proved by previous researches to enter the soil on both sides of the highway from vehicle mechanical wear and fuel consumption [11,62]. Our findings are consistent with their results. The Zn content in the study area did not increase significantly and was generally lower than the background value of Tibet in China [23]. Parts of the vehicle that are galvanized or contain zinc, such as fuel tanks and tires, may be a source of zinc contamination [63]. This may be due to the fact that there is little human activity other than the normal driving of vehicles in the study section, and there is little exposure and wear of galvanized parts. The research shows that the enrichment of heavy metals in the 4000 m area is more obvious in the highway disturbance area, and the overall content is higher, which may be related to the differences in traffic flow and vegetation types. Heavy metals might be transferred to the soil along the highway through airflow or pavement runoff [64,65]. In this study, the range of significant increase of heavy metal content along the highway was mainly limited to 25–50 m from the highway, which may be related to the location of the transect in this study. This study area is selected in the no man’s land and nature reserve without other human activities except roads, with few people and no grazing except for normal vehicle driving and road maintenance. Wildlife activity is also relatively low, which may lead to weak diffusion of heavy metal particles.

### 4.2. Effects of Environmental Factors on Soil Bacterial Community Structure at Different Altitudes

Therefore, the construction and traffic of the highway will significantly change the vegetation growth, soil physical, and chemical properties and content in grassland soil of heavy metal along the highway, thus affecting the diversity and structure of soil microbial communities. For example, Li et al. reported that the species composition of soil bacteria changed significantly during the degradation of vegetation [66]. Kang et al. studied the microbial community in an alpine wetland ecosystem and showed that human-induced pH changed microbial diversity and community structure in the upper soil layer as the main driving factors [67]. Underground coal mining has been reported to cause changes in soil conductivity and water content in sandy areas of western China, which can change the structure and diversity of soil microbial communities [68]. Research on the stressed soil near the QTH and Qinghai-Tibet Railway (QTR) proved heavy metals were important factors in the formation of bacterial community diversity [26]. In the Changbai mountain area, the disturbance of the highway was studied alongside the microbial diversity and communities in turf marsh soils, with the result that the microbial composition of turf marsh soils were mainly changed by road runoff and heavy metals emission [14]. In this study, the sequence data obtained indicate that the microbial communities differ significantly at the genus level.

In this study, the main dominant species at the phylum level are consistent with many previous studies (Figure 15A) [14,69,70]. In general, in disturbed or stressed soils, the abundance of *Proteobacteria* significantly decreased, while *Acidobacteria* abundance significantly increased compared with the undisturbed grassland [14,71]. In this study, this change was only consistent and significant in some areas, which may be related to the different degrees of environmental factors in different areas (Figure 15B). The ratio of *Proteobacteria* to *Acidobacteria* has been proved to be an indicator of changes in soil environmental conditions by several scholars [18]. This ratio between the road disturbed area and the control area was significantly different at three sites, with a significant increase at the 4000 m site and a significant decrease at the 5200 m site, indicating that the soil environmental conditions in the road-disturbed area have indeed changed in this study. He et al.’s research shows that the significant decrease of this ratio in the grassland affected by the highway is related to the fact that the altitude of the study area is much lower than 4000 m. This can be explained by geographical location and soil properties. *Alistipes* and *Barnesiella* are derived from faeces and are more abundant in unaffected areas. *Barnesiella* is in a similar situation.

At the genus level, the changes of different bacteria in this study showed a variety of patterns, which may be related to their characteristics. For example, *Flavisolibacter* was isolated from automotive air conditioners in previous studies [70]. Its significant enrichment occurred within 20 m from the road, which may be due to the movement of the vehicle. The *Blautia* abundance significantly increased in the control which was isolated from animals and humans [69,72]. This could be due to the fact that the vegetation in the unaffected area remained in the original state, the quality of grassland was high, and there were more wildlife activities. It could be seen that due to the impact of highway construction and operation, the activities of wildlife were reduced in the area near the highway, as were *Alistipes* and *Barnesiella*, which may also come from animal dung [73,74,75]. This was also confirmed by the increase in abundance in the unaffected areas. Abundance of *Microvirga*, *Skermanella* and *Crossiella* correlates with pH [76]. Due to the optimum pH (7.0) for growth, the abundance of those was negatively related to pH in the alkaline environment of the study area, which was also confirmed by the results of this research. Consistently in this study, a significant positive correlation was found between *Pseudomonas* abundance and N content [77]. Another study proved [78] that *Pseudomonas* abundance in the soil samples of the non-plant area was significantly enriched compared with the composite rhizosphere soil samples. Consistent with previous studies, the vegetation coverage of the 20 m treatment declined significantly, while *Pseudomonas* abundance significantly increased. Liu et al. reported that the *Pseudomonas* can promote the degradation of weathered diesel-oil pollutants to some extent. The increased abundance of *Pseudomonas* in the area near the highway in this study may be related to its adaptability to diesel pollutants caused by the highway. Consistent with previous studies, *Marmoricola* abundance correlated with soil carbon content positively [8]. There was a reported significant positive correlation between *Nocardioides* abundance and soil phosphorus content [79], which is consistent with the enrichment of soil-available phosphorus in this study. At 5200 m and 4600 m, soil-available phosphorus increased significantly near the highway. In addition, different bacteria responded differently to heavy metals, and those sensitive to toxicity decreased, while resistant bacteria could adapt to environmental changes and their relative abundance increased. For example, the relative abundance of *Flavobacterium*, *Gemmatimonas*, *Terrimonas*, *Nitrospira,* and *Bacillus* had increased, while *Barnesiella* and *Blastococcus’* relative abundance decreased in metal-enriched soil [14,80,81]. Our results are consistent with these observations. *Flavobacterium* and *Pseudomonas* were more significantly enriched in the disturbed area at the height of 4000 m. Accordingly, the content of heavy metals at 4000 m is relatively high. This indicated that heavy metals had an impact on the soil bacterial community structure. *Prevotella* 9, *Blautia*, *Marmoricola,* and *Barnesiella* were significantly enriched in the control group and negatively correlated with the concentration of heavy metals, which were sensitive to heavy metals. Consistent with the results of this study, *Marmoricola* abundance was shown to be significantly negatively correlated with soil heavy metal content [82]. *Gemmatimonas*, *Microvirga*, *Massilia*, *Sphingomonas,* and *Blastococcus* were significantly enriched in the highway disturbed area, and were positively correlated with heavy metals, indicating that they were heavy metal tolerant bacterium [83,84,85].

### 4.3. Changes in Soil Bacterial Functional Groups under the Influence of Highway

FAPROTAX analysis showed that the functional groups with significant changes in different sites were mainly concentrated in bacteria related to soil nitrogen cycling. Ammonia oxidation, nitrite oxidation, nitrification and other soil nitrifying bacteria in the disturbed area from the road showed an upward trend compared with the control. Ammonia oxidation by ammonia-oxidizing bacteria is the first step in nitrification and the rate limiting step [60]. The conversion of ecosystem types may have important potential impacts on soil microorganisms involved in ammonia oxidation, and there is a significantly positively correlation between the number of soil ammonia-oxidizing bacteria and soil NH_4_^+^-N content [86]. NH_4_^+^-N significantly affected the population composition of soil ammonia-oxidizing bacteria in previous studies [80]. The results showed that the content of soil ammonium nitrogen increased significantly after the moderate degradation stage of alpine meadow, and the soil ammonium accumulation was obvious in the later stage of grassland degradation. Consistent with previous studies, the number of aerobic nitrite oxidation bacterial at 5 m significantly increased compared with the control in the 4600 m site, which was similar to the change trend of soil ammonium nitrogen. The increase of soil ammonium nitrogen was the main reason for the enrichment of ammonia-oxidizing functional genes. Nitrite oxidation is the second step of nitrification. At the 5200 m and 4600 m sites, the number of nitrite-oxidizing bacteria increased significantly at 5 m sites compared with the control, which may be due to the accumulation of ammonium nitrogen in the degraded grassland affected by the highway, which promoted ammonia oxidation, thus providing a material basis for the nitrite oxidation process. However, the content of nitrite in the soil was not measured in this study, so the specific reasons for this process need further study.

In the process of soil denitrification, the bacterial communities related to nitrate reduction decreased significantly in the sites within 100 m from the road, which was opposite to the change of nitrification. Nitrate reduction and nitrite ammonification are two important processes of soil denitrification. Nitrate reduction is the first step of denitrification. Nitrate-reducing bacteria decreased significantly in the disturbed area of the highway, and their abundance was positively correlated with distance. There is a positive correlation between distance and nitrate bacteria. In the road disturbed area, the content of soil nitrate decreased, the material base of nitrate-reducing bacteria decreased, and the substrate of nitrate reduction reaction decreased, so the abundance of nitrate-reducing bacteria decreased. Nitrate bacteria relative abundance significantly correlated with altitude, and the same correlation occurs between nitrate-reducing bacteria and altitude. The reason is that with the increase of altitude, the oxygen content and the relative abundance of anaerobic nitrate-reducing bacteria increase, and the nitrate reduction reaction becomes more intense, consuming nitrate, therefore the content of nitrate decreases.

In the disturbed areas of 4600 m and 5200 m, the content of nitrate decreased and the content of ammonium nitrogen increased, indicating that soil denitrification was dominant. These might be due to the deterioration of soil aeration caused by the slight degradation of grassland, the decrease of vegetation coverage, the increase of soil compaction and the compression of soil pore structure [81], and then the promotion of the growth of anaerobic denitrifying bacteria. Firstly, because of the decrease of nitrate content, nitrate-reducing bacteria, and the reaction product nitrite and the nitrogen-fixing bacteria, nitrous oxide denitrifying bacteria increases, and part of nitrogen in soil enters the air in the form of nitrous oxide and nitrogen. This also confirms the results of the decrease of total soil nitrogen and AN content. Nitrogen-fixing bacteria decreased, but the content of ammonium nitrogen increased, indicating that the way of denitrification to produce ammonium nitrogen may be mainly through nitrite ammonification [87]. The decrease of SM proved it could significantly increase nitrite-oxidizing bacteria abundance, which also confirmed the significant increase of nitrite-ammonifying bacteria in this study. This may also be due to the deterioration of soil aeration, which reduces the performance of soil respiration, weakens nitrogen fixation, and promotes the production of anaerobic bacteria in the soil. According to Yu et al., in farmland soil, the abundance of denitrifying bacteria and nitrogen-fixing bacteria decreased significantly due to the decrease of nitrogen content after long-term nitrogen application [88]. It was further confirmed that the growth of denitrifying bacteria was limited by the decrease of soil nitrogen content. On the contrary, the nitrate and ammonium nitrogen decreased significantly at 4000 m, which may be due to the more serious grassland degradation and the overall decrease of nitrate and ammonium nitrogen. At the same time, denitrification-related bacteria also decreased in the disturbed area and significantly reduced. Combined with the results of vegetation index, it can be seen that the degradation degree of grassland in the highway disturbance area of the 4000 m site is higher, referring to the classification standards commonly used by predecessors [86], to the extent of moderate or severe degradation. Studies have shown that the soils C and N in severely degraded grassland are significantly reduced. The abundance of bacteria related to the nitrogen cycle showed a downward trend because of the lack of material basis for growth.

In addition, the chemoheterotrophic bacterial community showed a significant upward trend within 50 m from the road, and showed a consistent rule in the three areas, as the similar trend of previous studies. Sulfide respiration bacteria in the road disturbance area showed a downward trend compared with the control, and the lowest value (50 m) was found in the moderately degraded grassland soil, which was consistent with previous studies.

### 4.4. Relative Contribution of Vegetation, Soil, and Spatial Factors to Bacterial Community Structure in Different Altitudes

Mantel test results showed that biomass significantly affected the soil bacterial community structure (R = 0.241, *p* = 0.001). Among the soil factors, SOC, TN, AN, and NH4 were the main driving factors for the difference of soil bacterial community structure. AK and SM were the main driving factors for the difference of soil bacterial functional groups. This study found that soil nutrients are the most critical. As the most important nutrients for organisms, SOC and TN have been proved by many studies to be significantly related to soil microorganisms in different ecosystems [89,90]. Due to the anoxic soil conditions in the QTP, the decomposition of organic matter is slow. The lack of available nutrients, especially in areas disturbed by roads, limits the growth of bacteria. According to the traditional niche theory, the niches of many organisms may be related to available nitrogen, which causes impact on their coexistence in the ecosystem and changes the community structure [91]. Changes in TN, AN, and NH_4_ can lead to changes in niche size, which can alter bacterial diversity [92].

According to the results of redundancy analysis (RDA), altitude played an important role in affecting the soil bacterial community structure. The responses of soil bacterial communities to environmental factors at different altitudes were further discussed, and the main driving factors of soil bacterial communities varied with different altitudes. For example, in the 4000 m site, the content of Zn and Pb were important factors driving the change of the soil bacterial community. Consistently, the soil heavy metal content was higher at 4000 m. This shows that the heavy metal pollution is more serious in the 4000 m environment of the disturbed area. The toxicity of heavy metals leads to the change of the soil bacterial community structure. The results of the RDA analysis at different altitudes also proved that the distance from the road greatly affected the structure of soil bacterial communities. The community structure and functional groups of soil bacteria were significantly changed by the changes of grassland vegetation and soil in the disturbed area of the highway.

In general, the effects of the highway on soil bacterial community structure and functional groups in the disturbed area are multifaceted, which is the result of the joint action of vegetation and soil. The effects of soil-vegetation coupling on the bacterial community structure and function were greater than the effects of soil-vegetation coupling alone. This phenomenon has also been reflected in previous studies. This shows that the influence of the environment on the growth of soil bacteria is a very complex process. Road traffic has a significant impact on all aspects of environmental factors, and the resulting changes in soil bacterial communities need to be explained in many ways. Therefore, further research is of great need in this area.

### 4.5. Uncertainty and Perspectives

Our study describes above-ground vegetation, soil chemistry, and bacterial communities at different elevations along the QTH. In addition, the driving mechanism of microbial community diversity and community structure changes in the highway-disturbed area of the QTP explored.

Several shortcomings of this study need to be considered. First, there is no repetition in the same altitude environment; however, the results of the study at different altitudes are generally consistent and can be verified by each other, which does not affect the reliability of our conclusions. Secondly, due to legal and policy requirements, we have no way to obtain specific traffic flow of data in the sampling zone. This might also play an important role in the change of grassland vegetation and soil bacteria. By further measuring more comprehensive indicators related to the soil nitrogen cycle (such as nitrate, nitrite, etc.), the response mechanism of bacterial functional groups related to the soil nitrogen cycle can be more clearly expounded. These problems deserve further study.

The impact of highway construction and transportation on soil microbial diversity of the ecosystem along the highway has attracted more and more attention [93,94,95]. A recent study found that soil nitrogen was a key factor driving changes in soil microbial biomass and enzyme activity in cutting slopes [96]. Interactions between potentially toxic substances from vehicle emissions, roadside soils, and associated biota have also recently been reviewed [97]. Soil nutrients such as TOC and TN were found to be the most important variables affecting soil bacterial diversity and community structure along the QTH in our research which was significant for the research of the ecological restoration process along the plateau highway in the future.

## 5. Conclusions

Based on high-throughput sequencing, physical and chemical parameters, and statistical analysis, this study explored the effects of traffic and construction of the Qinghai-Tibet Highway on soil bacterial communities and diversity along the highway. The results showed that the road-related activities led to vegetation degradation, significantly changed the physical and chemical properties of soil, and caused heavy metal pollution. These environmental factors significantly affected the soil bacteria diversity and community structure. Soil organic carbon (SOC) and total nitrogen (TN) were the main factors driving the difference of the soil bacterial community structure in the disturbed area. The main factor of grassland disturbance along the Qinghai-Tibet Highway is caused by the decrease of soil-nutrient content. This disturbance shows a trend of increasing gradually from closer to farther distances. Therefore, in the restoration stage of the highway slope, it is very important to maintain the balance of soil nutrient supply for the restoration of the underground ecosystem function. The bacteria that showed heavy metal tolerance in the study may be explored for potential use in soil bioremediation in future studies. These results may provide guidance for the grassland ecosystem restoration along the highway and the selection of highway construction schemes in the Qinghai-Tibet region.

## Figures and Tables

**Figure 1 ijerph-18-13137-f001:**
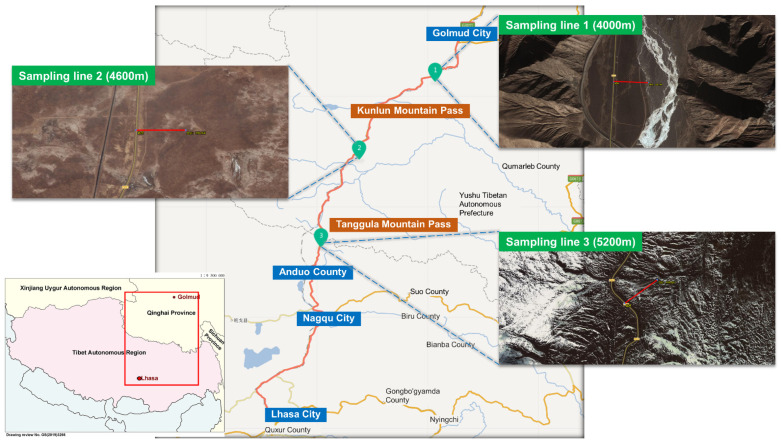
Sampling sites along the Qinghai–Tibet highway, China.

**Figure 2 ijerph-18-13137-f002:**
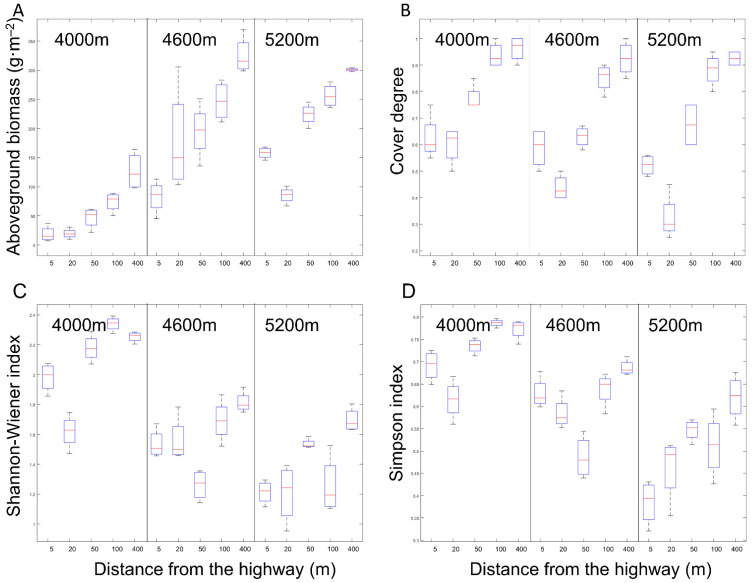
Comparison of (**A**) above-ground biomass, (**B**) plant cover degree, (**C**) plant simpson and (**D**) above-ground biomass between treatments at 5, 20, 50, 100 and 400 m from the curb G109 highways at altitudes of 4000 m (L), 4600 m (M) and 5200 m (H).

**Figure 3 ijerph-18-13137-f003:**
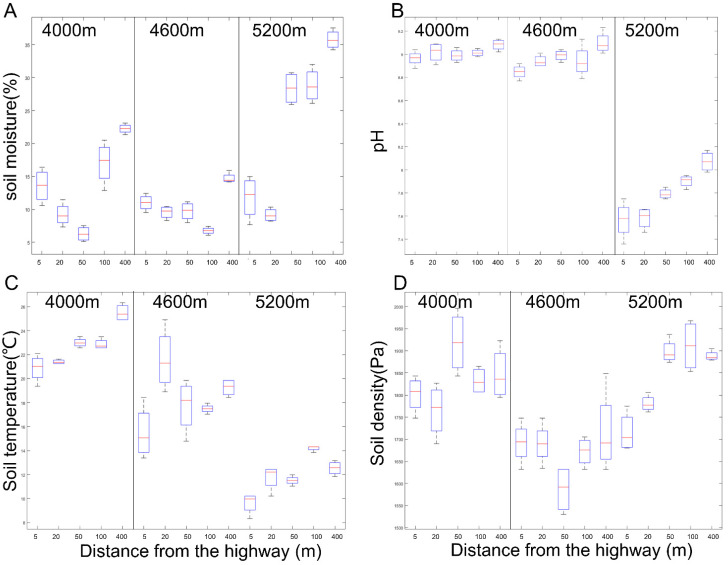
Comparison of soil: (**A**) soil moisture content, (**B**) pH, (**C**) soil temperature and (**D**) soil density between treatments at 5, 20, 50, 100, and 400 m from the curb G109 highways at altitudes of 4000 m (L), 4600 m (M), and 5200 m (H).

**Figure 4 ijerph-18-13137-f004:**
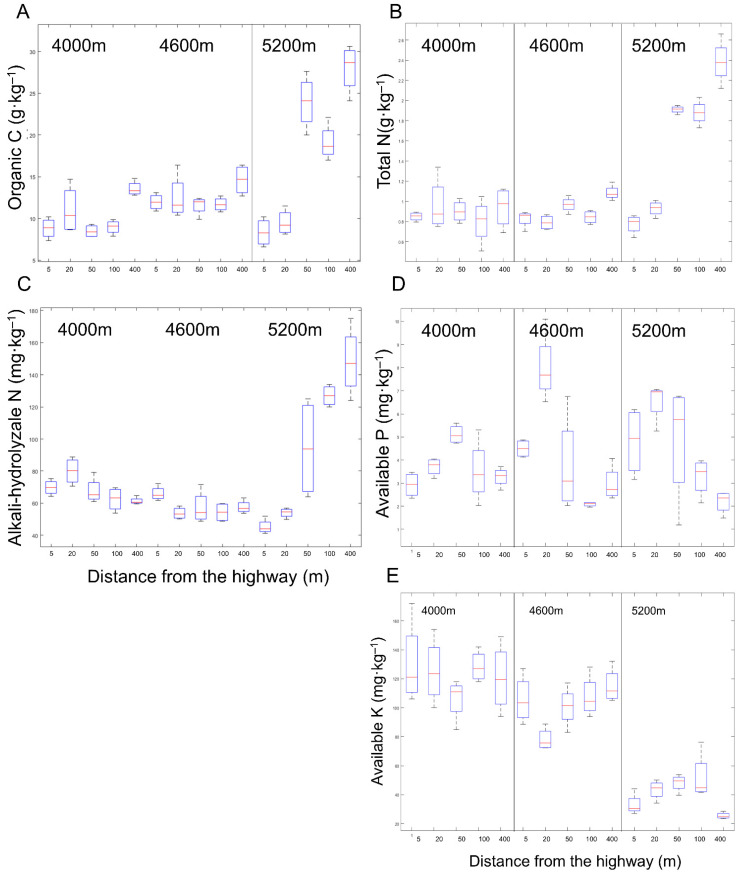
Comparison of soil: (**A**) organic carbon (SOC), (**B**) total nitrogen (TN), (**C**) alkali-hydrolyzable nitrogen (AN), (**D**) available phosphorus (AP), and (**E**) available potassium (AK) between treatments at 5, 20, 50, 100, and 400 m from the curb G109 highways at altitudes of 4000 m (L), 4600 m (M), and 5200 m (H).

**Figure 5 ijerph-18-13137-f005:**
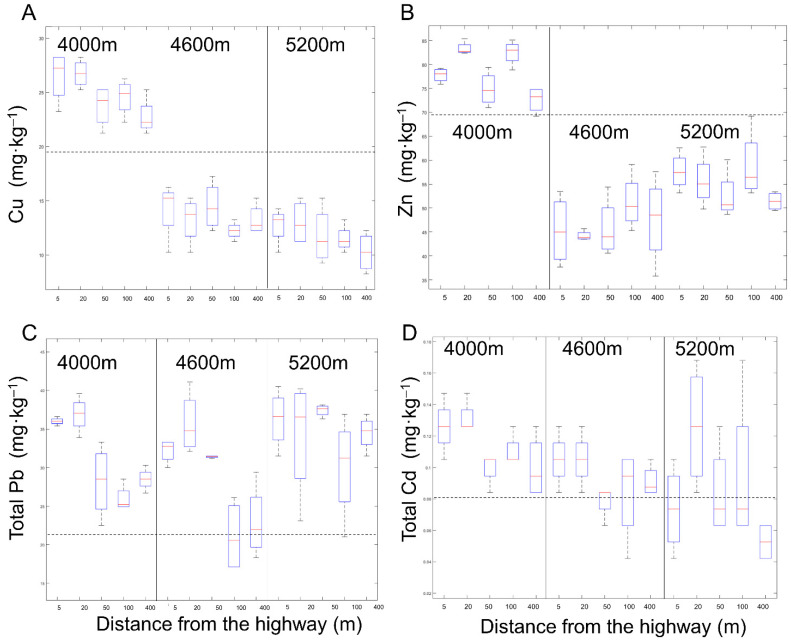
Concentrations of Cu (**A**), Zn (**B**), Pb (**C**), and Cd (**D**) in the 0–20 cm soil in sites 5, 20, 50, 100, and 400 m away from the G109 highway at altitudes of 4000 m (L), 4600 m (M), and 5200 m (H). The dotted line in the figure represents the background value of heavy metals in the Qinghai-Tibet Plateau.

**Figure 6 ijerph-18-13137-f006:**
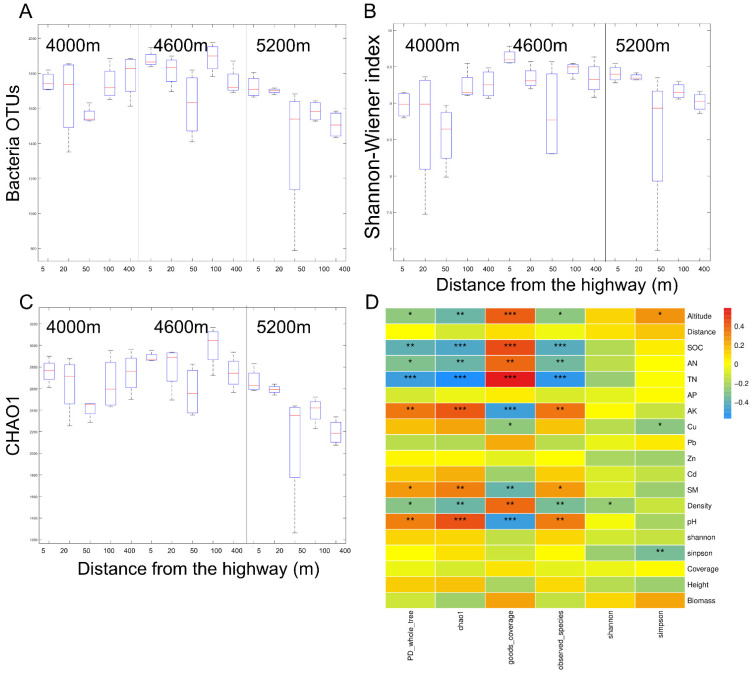
Box Diagram of (**A**) bacterial operational taxonomic unit (OTU), (**B**) bacterial Shannon and (**C**) bacterial CHAO1 in the 0–20 cm soil depths at 5, 20, 50, 100, and 400 m from G109 highway at altitudes of 4000 m (L), 4600 m, (M) and 5200 m (H). Pearson correlation heatmaps are based on bacterial alpha diversity estimators and environmental factors (**D**). * Means *p* < 0.05 for significance test; ** means *p* < 0.01 for significance test; *** means *p* < 0.001 for significance test.

**Figure 7 ijerph-18-13137-f007:**
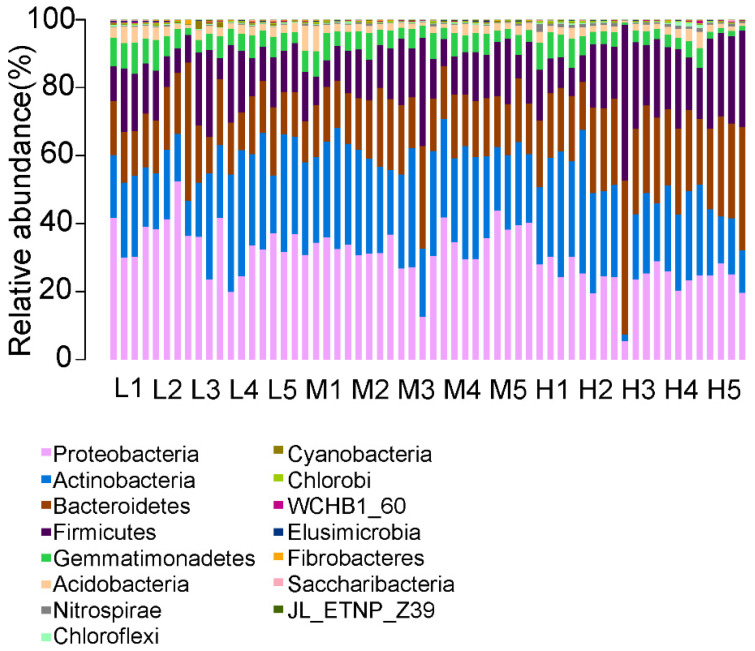
Chart of the relative abundance of the different levels of bacteria at the phylum level in the 0–20 cm depths soil at 5, 20, 50, 100, and 400 m from the curb G109 highways at altitudes of 4000 m, 4600 m, and 5200 m.

**Figure 8 ijerph-18-13137-f008:**
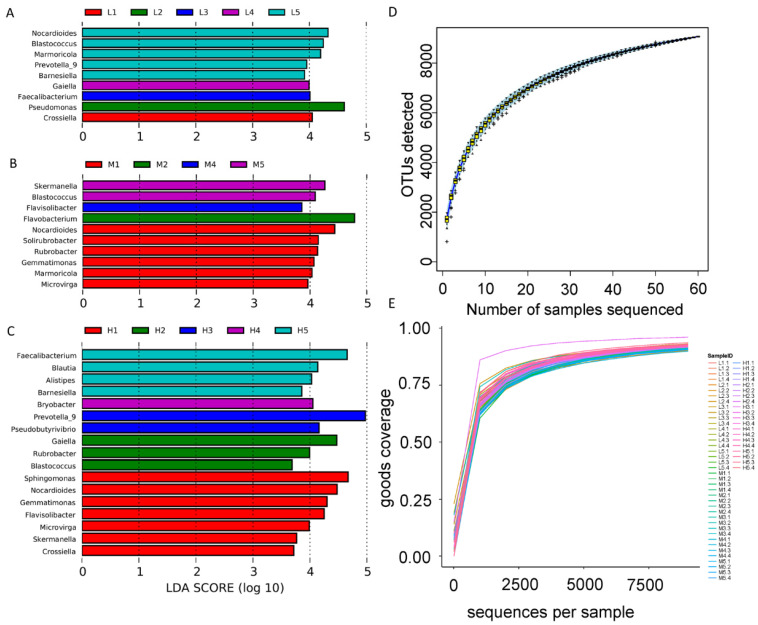
Linear discriminant analysis (LDA) of soil bacterial community in 3 sites of (**A**) 4000 m, (**B**) 4600 m, and (**C**) 5200 m. Rarefaction curve (**D**) and species accumulation curves (**E**) of the 60 soil samples.

**Figure 9 ijerph-18-13137-f009:**
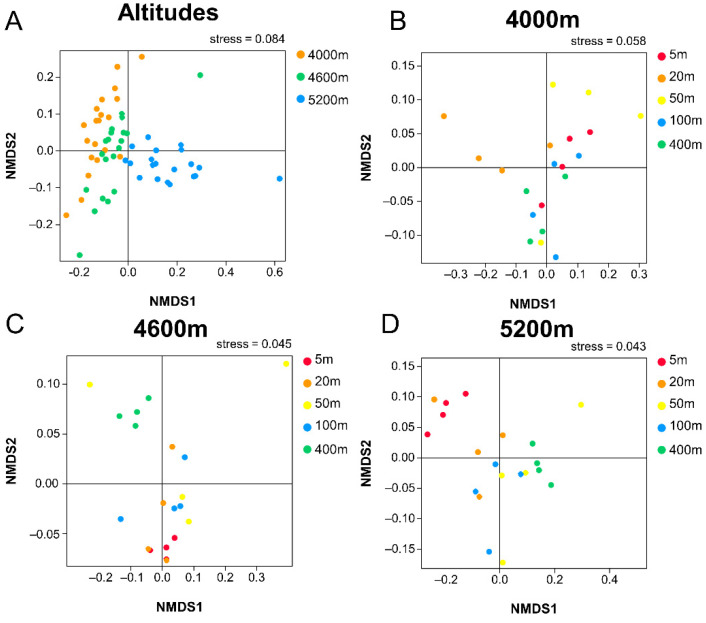
Non-metric multidimensional scaling (NMDS) analysis of (**A**) overall bacterial in 3 altitudes and bacterial in 3 sites of (**B**) 4000 m, (**C**) 4600 m, and (**D**) 5200 m.

**Figure 10 ijerph-18-13137-f010:**
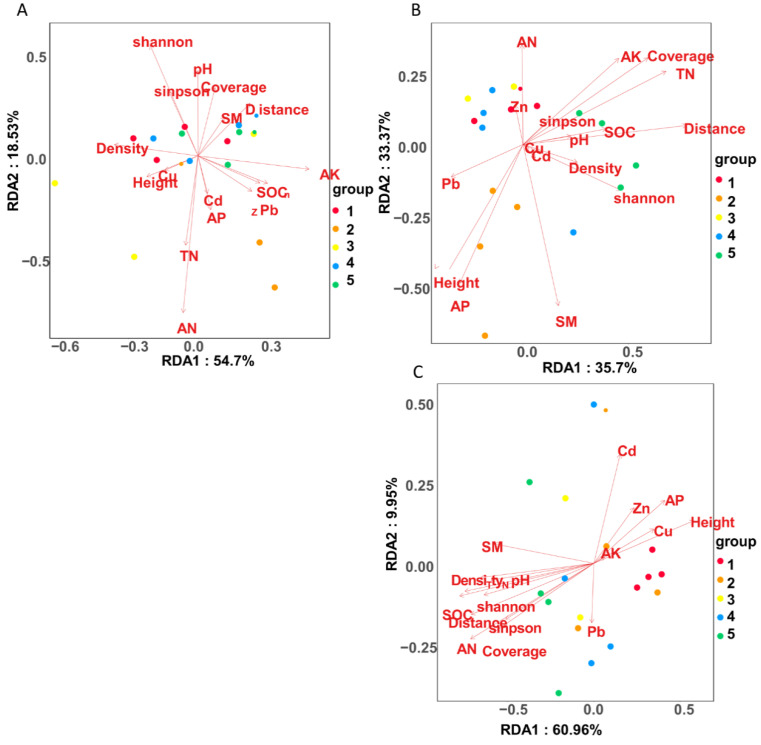
Ordination biplots of redundancy analysis (RDA) analysis of bacteria community structure, physical and chemical properties of soil and plant between treatments at 5, 20, 50, 100, and 400 m from the curb G109 highways at altitudes of (**A**) 4000 m, (**B**) 4600 m, and (**C**) 5200 m.

**Figure 11 ijerph-18-13137-f011:**
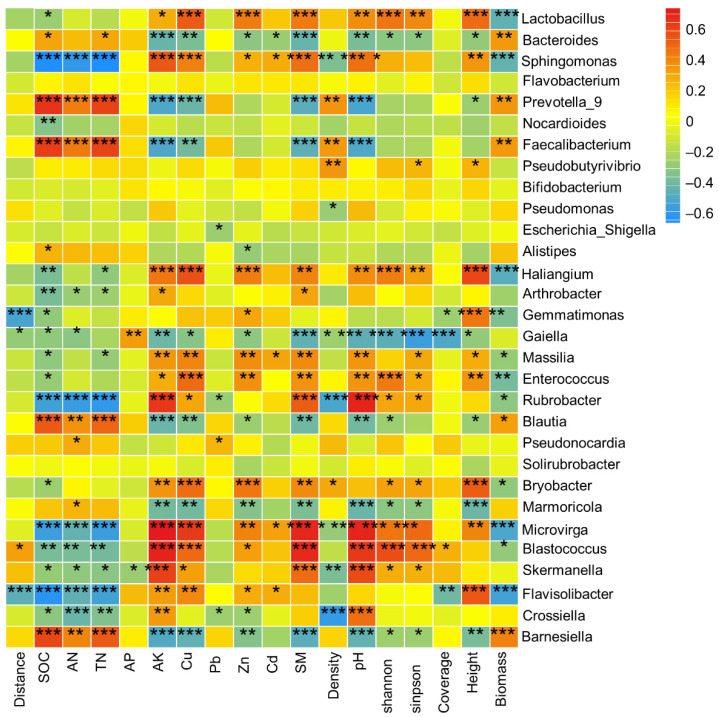
Pearson correlation heat map with correlation coefficient and significance levels based on the relative abundance of bacteria at the genus level and environmental factors. * Means *p* < 0.05 for significance test; ** means *p* < 0.01 for significance test; *** means *p* < 0.001 for significance test.

**Figure 12 ijerph-18-13137-f012:**
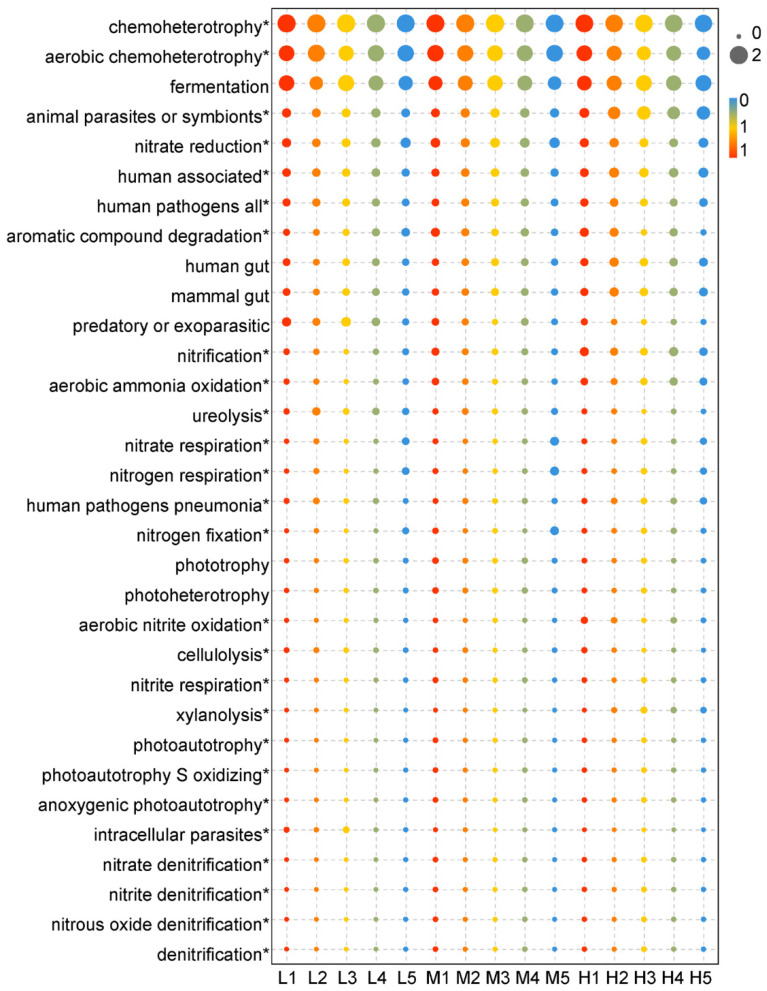
Ecological functional diversity of soil bacterial community in different altitudes and distances. * *p* < 0.05.

**Figure 13 ijerph-18-13137-f013:**
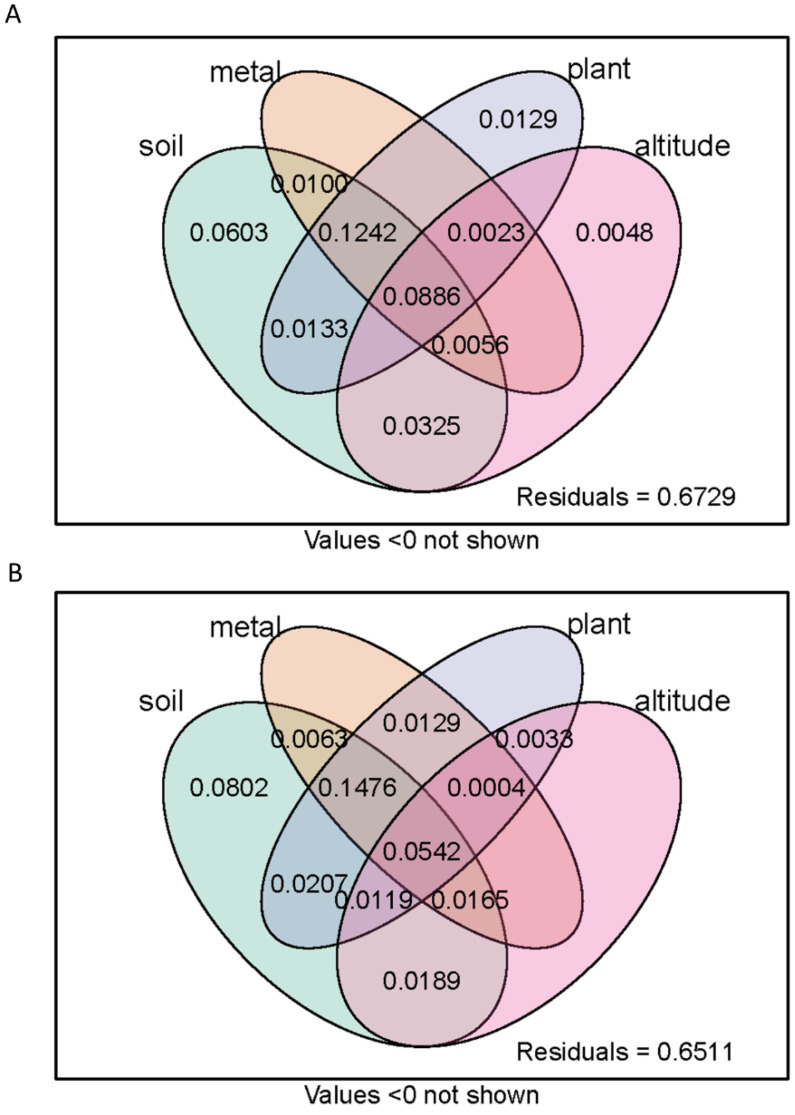
Variation partitioning analysis (VPA) which showed the relative proportions of total (**A**) bacterial and (**B**) functional diversity composition variations that can be explained by different types of environmental factors. The circles show the variation explained by each group of environmental factors alone.

**Figure 14 ijerph-18-13137-f014:**
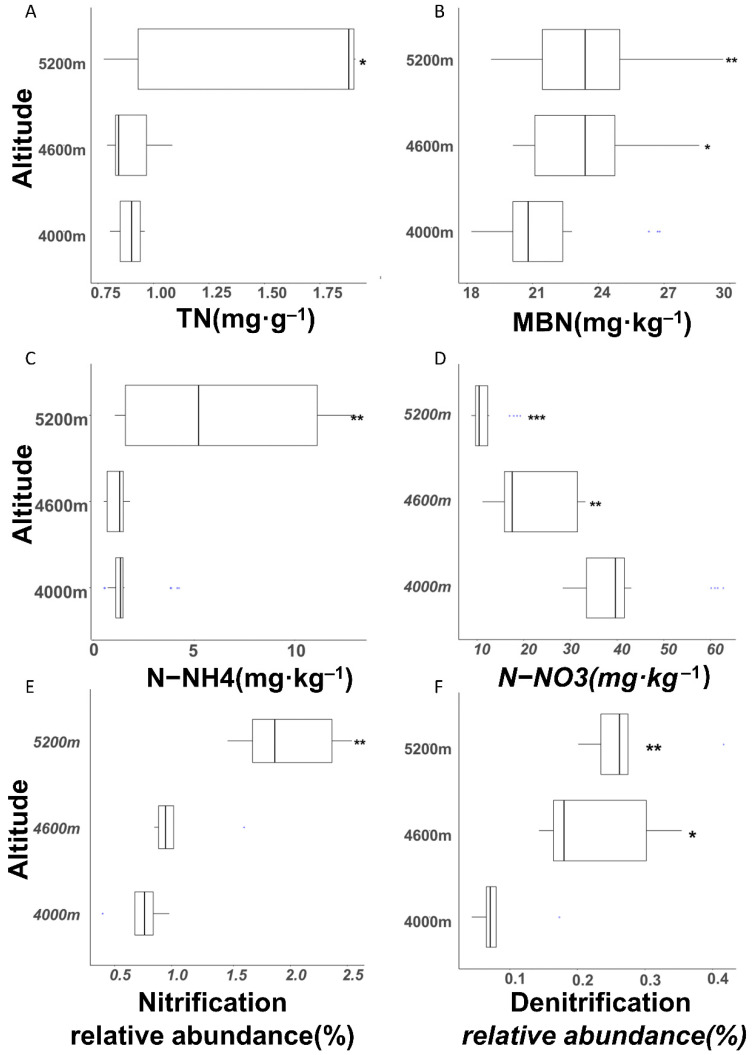
N cycle related indicators: (**A**) total soil nitrogen content, (**B**) soil microbial nitrogen content, (**C**) soil ammonia nitrogen N-NH_4_, (**D**) soil nitrate N-NO_3_, (**E**) nitrification bacteria relative content, (**F**) the relative content of denitrification bacteria at 4000 m (L), 4600 m (M), and 5200 m (H). * Means *p* < 0.05 for significance test; ** means *p* < 0.01 for significance test; *** means *p* < 0.001 for significance test.

**Figure 15 ijerph-18-13137-f015:**
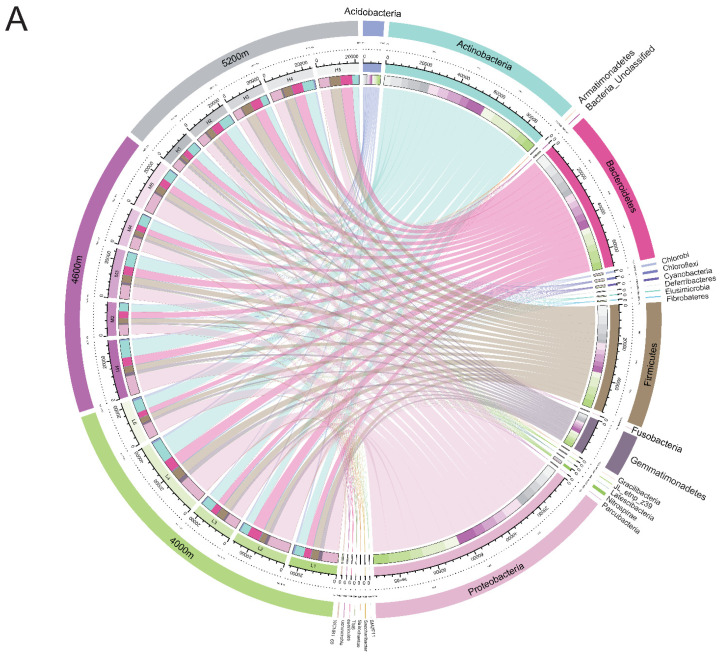
Circos circular plot of the bacterial community (**A**) at the phylum level and (**B**) at the genus level.

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
