# Peer review of "Effects of Elevation and Distance from Highway on the Abundance and Community Structure of Bacteria in Soil along Qinghai-Tibet Highway"

_ijerph, 2021, doi:10.3390/ijerph182413137_

Round 1

Reviewer 1 Report

Dear authors

This article provides new insights into the impact of a major highway on the abundance and community structure of soil bacteria. The manuscript is well written and the statistical analyses used are very well employed. I find it suitable for the journal, pointing some minor corrections before its possible acceptance:

  • A brief introduction to the Bacterial alpha Diversity Index would enhance the understanding of the subsequent analysis.
  • The scientific names of the species and/or genera of bacteria cited should be well written, all in italics. For example lines 327, 337, 396, and many more.

Nitrospirillum instead of Nitrospirillum, Pseudomonas instead of P seudomonas (remove the gap and put in italic), Lactobacillus, Enterococcus, Mycobacterium, Bradyrhizobium….

Line 137: DNA instead of dna

Line 144: The Walkey instead of Thewalkey

Lines 165, 188, 190: genes must be written correctly and in italics:  16S rRNA genes and 18S rRNA genes.

Section 3.6 Bacterial Community Diversity. The authors should review the selected functional groups because it is not well understood that metabolic concepts (type of metabolism) and other unrelated concepts are mixed. It is not understood why Animal Parasites and Animal Symbionts groups are included. You should consider that the same species of symbiont bacteria can also be opportunistic parasites on many occasions. Moreover, both of these functional groups of bacteria are chemoorganotrophs, so they would fall into that group. Please check these overlaps between the possible independent groups you cite in your paper.

On the other hand, it is not understood that a separate group of fermentative bacteria is made, when the fermentative metabolism is also chemoorganotrophic. Therefore, fermentatives would also belong to the chemoorganotrophic group. Please review this section and reconsider the criteria for the functional groups of bacteria made.

Line 427, 501, 578, 601, 617 .....: some capital letters should be revised.

Line 626: A full stop in the middle of the sentence.

Line 658: Nitrogen fixation is not a denitrification step. This is not correct, as nitrogen fixation involves the incorporation of N2 into the soil/water in the form of NH3 by nitrogen fixing bacteria, which is a combined nitrogen input to the soil and is therefore not denitrification. Denitrification is the anaerobic respiration of nitrites and nitrates, which removes nitrogen from the soil/water in the form of gases to the atmosphere.

Line 662: This is a suggestion: The term nitrate nitrogen is no longer in use. The term nitrate is more widely used and accepted.

Author Response

Response to Reviewer Comments

Point 1: A brief introduction to the Bacterial alpha Diversity Index would enhance the understanding of the subsequent analysis.

Response 1: Agree with the opinions of the reviewer. Upon request, a description of the bacterial alpha diversity index has been added to the new version of the manuscript. (New manuscript on line 258)

Point 2: The scientific names of the species and/or genera of bacteria cited should be well written, all in italics. For example lines 327, 337, 396, and many more.

Nitrospirillum instead of Nitrospirillum, Pseudomonas instead of P seudomonas (remove the gap and put in italic), Lactobacillus, Enterococcus, Mycobacterium, Bradyrhizobium….

Response 2: Agree with the reviewer's opinion. The scientific names of the species and genera of bacteria in the full text were proofread and modified and replaced as required. (Add new contributions on lines 15, 298, 303, 312, 551, 566, etc.)

Point 3: Line 137: DNA instead of dna

Response 3: We agree with the reviewer's comments and have been replaced. (Add new contribution on lines 133)

Point 4: Line 144: The Walkey instead of Thewalkey

Response 4: Agree with the opinions of the reviewer. Due to language issues, we have deleted this sentence in this revision.

Point 5: Lines 165, 188, 190: genes must be written correctly and in italics: 16S rRNA genes and 18S rRNA genes.

Response 5: We agree with the opinions of the reviewer, and the related sentences have been revised. (Add new contribution on lines 154, 156, 180)

Point 6: Bacterial Community Diversity. The authors should review the selected functional groups because it is not well understood that metabolic concepts (type of metabolism) and other unrelated concepts are mixed. It is not understood why Animal Parasites and Animal Symbionts groups are included. You should consider that the same species of symbiont bacteria can also be opportunistic parasites on many occasions. Moreover, both of these functional groups of bacteria are chemoorganotrophs, so they would fall into that group. Please check these overlaps between the possible independent groups you cite in your paper.

On the other hand, it is not understood whether a separate group of fermentative bacteria is made when fermentative metabolism is also chemoorganotrophic. Therefore, fermentatives would also belong to the chemoorganotrophic group. Please review this section and reconsider the criteria for the functional groups of bacteria made.

Response 6: In this study, FAPROTAX was used to predict the function of the bacterial community. The FAPROTAX method is a functional annotation database based on prokaryotic microbial taxonomy created by Louca et al. In 2016 to analyze the function of microbial communities(Louca et al., 2016). The method is based on the current prokaryotic functional annotation database manually collated from the literature of culturable bacteria, which contains more than 7600 functional annotations collected from more than 4600 prokaryotic microorganisms in 80 functional groups (such as nitrate respiration, methanogenesis, fermentation, plant pathogens). Louca et al. wrote a Python script to run microbial function prediction. The input file format includes the OTU classification tables generated by the Silva and Greengenes databases. Due to the complexity of microbial functions, it is possible that the same microorganism belongs to more than one group in the FAPROTAX analysis results. In recent years, the application of the FAPROTAX method in microbial function prediction has been common. There is also overlap between groups in the findings of these scholars(Liang et al., 2019, Orlofsky et al., Peng et al., 2021, Picazo et al., 2021, Xu et al., 2021). Recently, the reliability of FAPROTAX has also been discussed, showing that FAPROTAX can be used for rapid functional screening or grouping of 16S bacterial data in terrestrial ecosystems, and its performance can be improved by improving taxonomic and functional reference databases(Sansupa et al., 2021).

Point 7: Line 427, 501, 578, 601, 617 .....: some capital letters should be revised.

Response 7: Agree with the opinions of the reviewer. We have carefully checked similar questions throughout the manuscript. The letters have been proofread and revised. (Add new contribution on lines 199, 414 etc.)

Point 8: Line 626: A full stop in the middle of the sentence.

Response 8: The reviewer's comments have been agreed upon, and the full stop in the sentence has been deleted. (New manuscript on line 630)

Point 9: Line 658: Nitrogen fixation is not a denitrification step. This is not correct, as nitrogen fixation involves the incorporation of N2 into the soil/water in the form of NH3 by nitrogen fixing bacteria, which is a combined nitrogen input to the soil and is therefore not denitrification. Denitrification is the anaerobic respiration of nitrites and nitrates, which removes nitrogen from the soil/water in the form of gases to the atmosphere.

Response 9: Agree with the opinions of the reviewer. The errors have been corrected. (New manuscript on line 631)

Point 10: Line 662: This is a suggestion: The term nitrate nitrogen is no longer in use. The term nitrate is more widely used and accepted.

Response 10: Agree with the reviewer. The word "nitrate nitrogen" in the new manuscript has been revised to "nitrate".

Liang, S., Deng, J., Jiang, Y., Wu, S. & Zhu, W. (2019). Polish Journal of Environmental Studies 29.

Louca, S., Parfrey, L. W. & Doebeli, M. (2016). Science 353, 1272.

Orlofsky, E., Zabari, L., Bonito, G. & Masaphy, S. Environ. Microbiol., 12.

Peng, S. J., Hao, W. J., Li, Y. X., Wang, L., Sun, T. T., Zhao, J. M. & Dong, Z. J. (2021). Frontiers in Microbiology 12, 16.

Picazo, A., Villaescusa, J. A., Rochera, C., Miralles-Lorenzo, J., Quesada, A. & Camacho, A. (2021). Microorganisms 9, 22.

Sansupa, C., Wahdan, S. F. M., Hossen, S., Disayathanoowat, T., Wubet, T. & Purahong, W. (2021). Appl. Sci.-Basel 11, 18.

Xu, Z., Ma, Y., Zhang, L., Han, Y. & Luo, W. (2021). Science of The Total Environment 767, 144210.

Reviewer 2 Report

Dear Authors

The work "Effect mechanism of land consolidation on soil bacterial community" deals with an important topic concerning the impact of land use on soil microorganisms. In my opinion, the work was prepared very carefully and reliably. Properly selected tools used for description. The material is presented comprehensively.

I have only minor comments about the separation - methodology and the chapter with conclusions. In the part describing the methodology, it is difficult to find information on the frequency of sampling for analyzes (how many and in what period). In my opinion, the summary chapter does not have to be so detailed. The individual points in the applications contain detailed information provided earlier in the text. I propose to provide more general information here.

Author Response

Response to Reviewer Comments

Point 1: In the part describing the methodology, it is difficult to find information on the frequency of sampling for analyzes (how many and in what period).

Response 1: Agree with the reviewer's opinion. It should be explained that all sampling in this study was completed within 10 days, according to the distance between the different sites. The sampling time is between 12:00 to 14:00. Five plots were set for each site. Four quadrats were set for each plot for repetition, and 5 times of sampling were set in each quadrat for mixing. Thus, a total of 15 treatments were set and 60 soil samples were collected. Some of the relevant information has been explained in the new manuscript. (New manuscript on line 126)

Point 2: In my opinion, the summary chapter does not have to be so detailed. The individual points in the applications contain detailed information provided earlier in the text. I propose to provide more general information here.

Response 2: Agree with the reviewer's opinion. The conclusion part has been revised. The specific information that has appeared in the previous text has been deleted, and more general information has been added. (New manuscript on line 748)
